# Data Poisoning Attacks on Factorization-Based Collaborative Filtering

**Bo Li** *
Vanderbilt University
`bo.li.2@vanderbilt.edu`

**Yining Wang** *
Carnegie Mellon University
`ynwang.yining@gmail.com`

**Aarti Singh**
Carnegie Mellon University
`aarti@cs.cmu.edu`

**Yevgeniy Vorobeychik**
Vanderbilt University
`yevgeniy.vorobeychik@vanderbilt.edu`

## Abstract

Recommendation and collaborative filtering systems are important in modern information and e-commerce applications. As these systems are becoming increasingly popular in the industry, their outputs could affect business decision making, introducing incentives for an adversarial party to compromise the availability or integrity of such systems. We introduce a data poisoning attack on collaborative filtering systems. We demonstrate how a powerful attacker with full knowledge of the learner can generate malicious data so as to maximize his/her malicious objectives, while at the same time mimicking normal user behavior to avoid being detected. While the complete knowledge assumption seems extreme, it enables a robust assessment of the vulnerability of collaborative filtering schemes to highly motivated attacks. We present efficient solutions for two popular factorization-based collaborative filtering algorithms: the *alternative minimization* formulation and the *nuclear norm minimization* method. Finally, we test the effectiveness of our proposed algorithms on real-world data and discuss potential defensive strategies.

## 1 Introduction

Recommendation systems have emerged as a crucial feature of many electronic commerce systems. In machine learning such problems are usually referred to as *collaborative filtering* or *matrix completion*, where the known users' preferences are abstracted into an incomplete user-by-item matrix, and the goal is to complete the matrix and subsequently make new item recommendations for each user. Existing approaches in the literature include nearest-neighbor methods, where a user's (item's) preference is determined by other users (items) with similar profiles [1], and factorization-based methods where the incomplete preference matrix is assumed to be approximately low-rank [2, 3].

As recommendation systems play an ever increasing role in current information and e-commerce systems, they are susceptible to a risk of being maliciously attacked. One particular form of attacks is called *data poisoning*, in which a malicious party creates dummy (malicious) users in a recommendation system with carefully chosen item preferences (i.e., data) such that the effectiveness or credibility of the system is maximally degraded. For example, an attacker might attempt to make recommendations that are as different as possible from those that would otherwise be made by the recommendation system. In another, more subtle, example, the attacker is associated with the producer of a specific movie or product, who may wish to increase or decrease the popularity of a certain item. In both cases, the credibility of a recommendation system is harmed by the malicious activities, which could lead to significant economic loss. Due to the open nature of recommendation

systems and their reliance on user-specified judgments for building profiles, various forms of attacks are possible and have been discussed, such as the random attack and random product push/nuke attack [4, 5]. However, these attacks are not formally analyzed and cannot be optimized according to specific collaborative filtering algorithms. As it is not difficult for attackers to determine the defender's filtering algorithm or even its parameters settings (e.g., through insider attacks), this can lead one to significantly under-estimate the attacker's ability and result in substantial loss.

We present a systematic approach to computing near-optimal data poisoning attacks for factorization-based collaborative filtering/recommendation models. We assume a highly motivated attacker with knowledge of both the learning algorithms and parameters of the learner following the Kerckhoffs' principle to ensure reliable vulnerability analysis in the worst case. We focus on two most popular algorithms: *alternating minimization* [6] and *nuclear norm minimization* [3]. Our main contributions are as follows:

- **Comprehensive characterization of attacker utilities:** We characterize several attacker utilities, which include *availability attacks*, where prediction error is increased, and *integrity attacks*, where item-specific objectives are considered. Optimal attack strategies for all utilities can be computed under a unified optimization framework.
- **Novel gradient computations:** Building upon existing gradient-based data poisoning frameworks [7, 8, 9], we develop novel methods for gradient computation based on first-order KKT conditions for two widely used algorithms: alternating minimization [6] and nuclear norm minimization [2]. The resulting derivations are highly non-trivial; in addition, to our knowledge this work is the first to give systematic data poisoning attacks for problems involving non-smooth nuclear norm type objectives.
- **Mimicking normal user behaviors:** For data poisoning attacks, most prior work focuses on maximizing attacker's utility. A less investigated problem is how to synthesize malicious data points that are hard for a defender to detect. In this paper we provide a novel technique based on *stochastic gradient Langevin dynamics* optimization [10] to produce malicious users that mimic normal user behaviors in order to avoid detection, while achieving attack objectives.

**Related Work:** There has been extensive prior research concerning the security of machine learning algorithms [11, 12, 13, 14, 15]. Biggio et al. pioneered the research of optimizing malicious data-driven attacks for kernel-based learning algorithms such as SVM [16]. The key optimization technique is to approximately compute implicit gradients of the solution of an optimization problem based on first-order KKT conditions. Similar techniques were later generalized to optimize data poisoning attacks for several other important learning algorithms, such as Lasso regression [7], topic modeling [8], and autoregressive models [17]. The reader may refer to [9] for a general algorithmic framework of the abovementioned methods.

In terms of collaborative filtering/matrix completion, there is another line of established research that focuses on *robust matrix completion*, in which a small portion of elements or rows in the underlying low-rank matrix is assumed to be arbitrarily perturbed [18, 19, 20, 21]. Specifically, the stability of alternating minimization solutions was analyzed with respect to malicious data manipulations in [22]. However, [22] assumes a globally optimal solution of alternating minimization can be obtained, which is rarely true in practice.

## 2 Preliminaries

We first set up the collaborative filtering/matrix completion problem and give an overview of existing low-rank factorization based approaches. Let $\mathbf{M} \in \mathbb{R}^{m \times n}$ be a data matrix consisting of $m$ rows and $n$ columns. $\mathbf{M}_{ij}$ for $i \in [m]$ and $j \in [n]$ would then correspond to the rating the $i$th user gives for the $j$th item. We use $\Omega = \{(i, j) : \mathbf{M}_{ij} \text{ is observed}\}$ to denote all observable entries in $\mathbf{M}$ and assume that $|\Omega| \ll mn$. We also use $\Omega_i \subseteq [n]$ and $\Omega'_j \subseteq [m]$ for columns (rows) that are observable at the $i$th row ($j$th column). The goal of collaborative filtering (also referred to as *matrix completion* in the statistical learning literature [2]) is then to recover the complete matrix $\mathbf{M}$ from few observations $\mathbf{M}_\Omega$.

The matrix completion problem is in general ill-posed as it is impossible to complete an arbitrary matrix with partial observations. As a result, additional assumptions are imposed on the underlying data matrix $\mathbf{M}$. One standard assumption is that $\mathbf{M}$ is very close to an $m \times n$ rank-$k$ matrix with

$k \ll \min(m,n)$. Under such assumptions, the complete matrix $\mathbf{M}$ can be recovered by solving the following optimization problem:

$$\min_{\mathbf{X} \in \mathbb{R}^{m \times n}} \|\mathcal{R}_\Omega(\mathbf{M} - \mathbf{X})\|_F^2, \quad s.t. \ \ \mathrm{rank}(\mathbf{X}) \leq k, \tag{1}$$

where $\|\mathbf{A}\|_F^2 = \sum_{i,j} \mathbf{A}_{ij}^2$ denotes the squared Frobenious norm of matrix $\mathbf{A}$ and $[\mathcal{R}_\Omega(\mathbf{A})]_{ij}$ equals $\mathbf{A}_{ij}$ if $(i,j) \in \Omega$ and 0 otherwise. Unfortunately, the feasible set in Eq. (1) is non-convex, making the optimimzation problem difficult to solve. There has been an extensive prior literature on approximately solving Eq. (1) and/or its surrogates that lead to two standard approaches: alternating minimization and nuclear norm minimization. For the first approach, one considers the following problem:

$$\min_{\mathbf{U} \in \mathbb{R}^{m \times k}, \mathbf{V} \in \mathbb{R}^{n \times k}} \left\{ \|\mathcal{R}_\Omega(\mathbf{M} - \mathbf{U}\mathbf{V}^\top)\|_F^2 + 2\lambda_U \|\mathbf{U}\|_F^2 + 2\lambda_V \|\mathbf{V}\|_F^2 \right\}. \tag{2}$$

Eq. (2) is equivalent to Eq. (1) when $\lambda_U = \lambda_V = 0$. In practice people usually set both regularization parameters $\lambda_U$ and $\lambda_V$ to be small positive constants in order to avoid large entries in the completed matrix and also improve convergence. Since Eq. (2) is bi-convex in $\mathbf{U}$ and $\mathbf{V}$, an *alternating minimization* procedure can be applied. Alternatively, one solves a *nuclear-norm minimization* problem

$$\min_{\mathbf{X} \in \mathbb{R}^{m \times n}} \|\mathcal{R}_\Omega(\mathbf{M} - \mathbf{X})\|_F^2 + 2\lambda \|\mathbf{X}\|_*, \tag{3}$$

where $\lambda > 0$ is a regularization parameter and $\|\mathbf{X}\|_* = \sum_{i=1}^{\mathrm{rank}(\mathbf{X})} |\sigma_i(\mathbf{X})|$ is the nuclear norm of $\mathbf{X}$, which acts as a convex surrogate of the rank function. Eq. (3) is a convex optimization function and can be solved using an iterative singular value thresholding algorithm [3]. It can be shown that both methods in Eq. (2) and (3) provably approximate the true underlying data matrix $\mathbf{M}$ under certain conditions [6, 2].

## 3   The Attack Model

In this section we describe the data poisoning attack model considered in this paper. For a data matrix consisting of $m$ users and $n$ items, the attacker is capable of adding $\alpha m$ malicious users to the training data matrix, and each malicious user is allowed to report his/her preference on at most $B$ items with each preference bounded in the range $[-\Lambda, \Lambda]$.

Before proceeding to describe the attacker's goals, we first introduce some notation to facilitate presentation. We use $\mathbf{M} \in \mathbb{R}^{m \times n}$ to denote the original data matrix and $\widetilde{\mathbf{M}} \in \mathbb{R}^{m' \times n}$ to denote the data matrix of all $m' = \alpha m$ malicious users. Let $\widetilde{\Omega}$ be the set of non-zero entries in $\widetilde{\mathbf{M}}$ and $\widetilde{\Omega}_i \subseteq [n]$ be all items that the $i$th malicious user rated. According to our attack models, $|\widetilde{\Omega}_i| \leq B$ for every $i \in \{1, \cdots, m'\}$ and $\|\widetilde{\mathbf{M}}\|_{\max} = \max |\widetilde{\mathbf{M}}_{ij}| \leq \Lambda$. Let $\mathbf{\Theta}_\lambda(\widetilde{\mathbf{M}}; \mathbf{M})$ be the optimal solution computed jointly on the original and poisoned data matrices $(\widetilde{\mathbf{M}}; \mathbf{M})$ using regularization parameters $\lambda$. For example, Eq. (2) becomes

$$\mathbf{\Theta}_\lambda(\widetilde{\mathbf{M}}; \mathbf{M}) = \arg\min_{\mathbf{U}, \widetilde{\mathbf{U}}, \mathbf{V}} \|\mathcal{R}_\Omega(\mathbf{M} - \mathbf{U}\mathbf{V}^\top)\|_F^2 + \|\mathcal{R}_{\widetilde{\Omega}}(\widetilde{\mathbf{M}} - \widetilde{\mathbf{U}}\mathbf{V}^\top)\|_F^2 + 2\lambda_U(\|\mathbf{U}\|_F^2 + \|\widetilde{\mathbf{U}}\|_F^2) + 2\lambda_V \|\mathbf{V}\|_F^2 \tag{4}$$

where the resulting $\mathbf{\Theta}$ consists of low-rank latent factors $\mathbf{U}, \widetilde{\mathbf{U}}$ for normal and malicious users as well as $\mathbf{V}$ for items. Simiarly, for the nuclear norm minimization formulation in Eq. (3), we have

$$\mathbf{\Theta}_\lambda(\widetilde{\mathbf{M}}; \mathbf{M}) = \arg\min_{\mathbf{X}, \widetilde{\mathbf{X}}} \|\mathcal{R}_\Omega(\mathbf{M} - \mathbf{X})\|_F^2 + \|\mathcal{R}_{\widetilde{\Omega}}(\widetilde{\mathbf{M}} - \widetilde{\mathbf{X}})\|_F^2 + 2\lambda \|(\mathbf{X}; \widetilde{\mathbf{X}})\|_*, \tag{5}$$

where $\mathbf{\Theta} = (\mathbf{X}, \widetilde{\mathbf{X}})$. Let $\widehat{\mathbf{M}}(\mathbf{\Theta})$ be the matrix estimated from learnt model $\mathbf{\Theta}$. For example, for Eq. (4) we have $\widehat{\mathbf{M}}(\mathbf{\Theta}) = \mathbf{U}\mathbf{V}^\top$ and for Eq. (5) we have $\widehat{\mathbf{M}}(\mathbf{\Theta}) = \mathbf{X}$. The goal of the attacker is to find optimal malicious users $\widetilde{\mathbf{M}}^*$ such that

$$\widetilde{\mathbf{M}}^* \in \mathrm{argmax}_{\widetilde{\mathbf{M}} \in \mathbb{M}} R(\widehat{\mathbf{M}}(\mathbf{\Theta}_\lambda(\widetilde{\mathbf{M}}; \mathbf{M})), \mathbf{M}). \tag{6}$$

Here $\mathbb{M} = \{\widetilde{\mathbf{M}} \in \mathbb{R}^{m' \times n} : |\widetilde{\Omega}_i| \leq B, \|\widetilde{\mathbf{M}}\|_{\max} \leq \Lambda\}$ is the set of all feasible poisoning attacks discussed earlier in this section and $R(\widehat{\mathbf{M}}, \mathbf{M})$ denotes the attacker's utility for diverting the collaborative filtering algorithm to predict $\widehat{\mathbf{M}}$ on an original data set $\mathbf{M}$, with the help of few malicious users $\widetilde{\mathbf{M}}$. Below we list several typical attacker utilities:

**Availability attack** the attacker wants to maximize the error of the collaborative filtering system, and eventually render the system useless. Suppose $\overline{\mathbf{M}}$ is the prediction of the collaborative filtering system without data poisoning attacks.[2] The utility function is then defined as the total amount of perturbation of predictions between $\overline{\mathbf{M}}$ and $\widehat{\mathbf{M}}$ (predictions after poisoning attacks) on unseen entries $\Omega^C$:

$$R^{\mathrm{av}}(\widehat{\mathbf{M}}, \mathbf{M}) = \|\mathcal{R}_{\Omega^C}(\widehat{\mathbf{M}} - \overline{\mathbf{M}})\|_F^2. \tag{7}$$

**Integrity attack** in this model the attacker wants to boost (or reduce) the popularity of a (subset) of items. Suppose $J_0 \subseteq [n]$ is the subset of items the attacker is interested in and $w : J_0 \to \mathbb{R}$ is a pre-specified weight vector by the attacker. The utility function is

$$R^{\mathrm{in}}_{J_0, w}(\widehat{\mathbf{M}}, \mathbf{M}) = \sum_{i=1}^{m} \sum_{j \in J_0} w(j) \widehat{\mathbf{M}}_{ij}. \tag{8}$$

**Hybrid attack** a hybrid loss function can also be defined:

$$R^{\mathrm{hybrid}}_{J_0, w, \mu}(\widehat{\mathbf{M}}, \mathbf{M}) = \mu_1 R^{\mathrm{av}}_{J_0, w}(\widehat{\mathbf{M}}, \mathbf{M}) + \mu_2 R^{\mathrm{in}}(\widehat{\mathbf{M}}, \mathbf{M}), \tag{9}$$

where $\mu = (\mu_1, \mu_2)$ are coefficients that trade off the availability and integrity attack objectives. In addition, $\mu_1$ could be negative, which models the case when the attacker wants to leave a "light trace": the attacker wants to make his item more popular while making the other recommendations of the system less perturbed to avoid detection.

## 4 Computing Optimal Attack Strategies

We describe practical algorithms to solve the optimization problem in Eq. (6) for optimal attack strategy $\widetilde{\mathbf{M}}^*$ that maximizes the attacker's utility. We first consider the alternating minimization formulation in Eq. (4) and derive a projected gradient ascent method that solves for the corresponding optimal attack strategy. Similar derivations are then extended to the nuclear norm minimization formulation in Eq. (5). Finally, we discuss how to design malicious users that mimic normal user behavior in order to avoid detection.

### 4.1 Attacking Alternating Minimization

We use the *projected gradient ascent* (PGA) method for solving the optimization problem in Eq. (6) with respect to the alternating minimization formulation in Eq. (4): in iteration $t$ we update $\widetilde{\mathbf{M}}^{(t)}$ as follows:

$$\widetilde{\mathbf{M}}^{(t+1)} = \mathrm{Proj}_{\mathbb{M}}\left(\widetilde{\mathbf{M}}^{(t)} + s_t \cdot \nabla_{\widetilde{\mathbf{M}}} R(\widehat{\mathbf{M}}, \mathbf{M})\right), \tag{10}$$

where $\mathrm{Proj}_{\mathbb{M}}(\cdot)$ is the projection operator onto the feasible region $\mathbb{M}$ and $s_t$ is the step size in iteration $t$. Note that the estimated matrix $\widehat{\mathbf{M}}$ depends on the model $\boldsymbol{\Theta}_\lambda(\widetilde{\mathbf{M}}; \mathbf{M})$ learnt on the joint data matrix, which further depends on the malicious users $\widetilde{\mathbf{M}}$. Since the constraint set $\mathbb{M}$ is highly non-convex, we generate $B$ items uniformly at random for each malicious user to rate. The $\mathrm{Proj}_{\mathbb{M}}(\cdot)$ operator then reduces to projecting each malicious users' rating vector onto an $\ell_\infty$ ball of diameter $\Lambda$, which can be easily evaluated by truncating all entries in $\widetilde{\mathbf{M}}$ at the level of $\pm\Lambda$.

We next show how to (approximately) compute $\nabla_{\widetilde{\mathbf{M}}} R(\widehat{\mathbf{M}}, \mathbf{M})$. This is challenging because one of the arguments in the loss function involves an implicit optimization problem. We first apply chain rule to arrive at

$$\nabla_{\widetilde{\mathbf{M}}} R(\widehat{\mathbf{M}}, \mathbf{M}) = \nabla_{\widetilde{\mathbf{M}}} \boldsymbol{\Theta}_\lambda(\widetilde{\mathbf{M}}; \mathbf{M}) \nabla_{\boldsymbol{\Theta}} R(\widehat{\mathbf{M}}, \mathbf{M}). \tag{11}$$

The second gradient (with respect to $\boldsymbol{\Theta}$) is easy to evaluate, as all loss functions mentioned in the previous section are smooth and differentiable. Detailed derivation of $\nabla_{\boldsymbol{\Theta}} R(\widehat{\mathbf{M}}, \mathbf{M})$ is deferred to Appendix A. On the other hand, the first gradient term term is much harder to evaluate because $\boldsymbol{\Theta}_\lambda(\cdot)$ is an optimization procedure. Inspired by [7, 8, 9], we exploit the KKT conditions of the optimization problem $\boldsymbol{\Theta}_\lambda(\cdot)$ to approximately compute $\nabla_{\widetilde{\mathbf{M}}} \boldsymbol{\Theta}_\lambda(\widetilde{\mathbf{M}}; \mathbf{M})$. More specifically, the optimal solution $\boldsymbol{\Theta} = (\mathbf{U}, \widetilde{\mathbf{U}}, \mathbf{V})$ of Eq. (4) satisfies

$$\lambda_U \boldsymbol{u}_i = \sum_{j \in \Omega_i} (\mathbf{M}_{ij} - \boldsymbol{u}_i^\top \boldsymbol{v}_j) \boldsymbol{v}_j;$$

**Algorithm 1** Optimizing $\widetilde{\mathbf{M}}$ via PGA

---

1: **Input**: Original partially observed $m \times n$ data matrix $\mathbf{M}$, algorithm regularization parameter $\lambda$, attack budget parameters $\alpha$, $B$ and $\Lambda$, attacker's utility function $R$, step size $\{s_t\}_{t=1}^{\infty}$.
2: **Initialization**: random $\widetilde{\mathbf{M}}^{(0)} \in \mathbb{M}$ with both ratings and rated items uniformly sampled at random; $t = 0$.
3: **while** $\widetilde{\mathbf{M}}^{(t)}$ does not converge **do**
4:     Compute the optimal solution $\boldsymbol{\Theta}_\lambda(\widetilde{\mathbf{M}}^{(t)}; \mathbf{M})$.
5:     Compute gradient $\nabla_{\widetilde{\mathbf{M}}} R(\widehat{\mathbf{M}}, \mathbf{M})$ using Eq. (10).
6:     Update: $\widehat{\mathbf{M}}^{(t+1)} = \mathrm{Proj}_{\mathbb{M}}(\widehat{\mathbf{M}}^{(t)} + s_t \nabla_{\widetilde{\mathbf{M}}} R)$.
7:     $t \leftarrow t + 1$.
8: **end while**
9: **Output**: $m' \times n$ malicious matrix $\widetilde{\mathbf{M}}^{(t)}$.

---

$$\lambda_U \tilde{\boldsymbol{u}}_i = \sum_{j \in \widetilde{\Omega}_i} (\widetilde{\mathbf{M}}_{ij} - \tilde{\boldsymbol{u}}_i^\top \boldsymbol{v}_j) \boldsymbol{v}_j;$$

$$\lambda_V \boldsymbol{v}_j = \sum_{i \in \Omega'_j} (\mathbf{M}_{ij} - \boldsymbol{u}_i^\top \boldsymbol{v}_j) \boldsymbol{u}_i + \sum_{i \in \widetilde{\Omega}'_j} (\widetilde{\mathbf{M}}_{ij} - \tilde{\boldsymbol{u}}_i^\top \boldsymbol{v}_j) \tilde{\boldsymbol{u}}_i,$$

where $\boldsymbol{u}_i, \tilde{\boldsymbol{u}}_i$ are the $i$th rows (of dimension $k$) in $\mathbf{U}$ or $\widetilde{\mathbf{U}}$ and $\boldsymbol{v}_j$ is the $j$th row (also of dimension $k$) in $\mathbf{V}$. Subsequently, $\{\boldsymbol{u}_i, \tilde{\boldsymbol{u}}_i, \boldsymbol{v}_j\}$ can be expressed as functions of the original and malicious data matrices $\mathbf{M}$ and $\widetilde{\mathbf{M}}$. Using the fact that $(\boldsymbol{a}^\top \boldsymbol{x})\boldsymbol{a} = (\boldsymbol{a}\boldsymbol{a}^\top)\boldsymbol{x}$ and $\mathbf{M}$ does not change with $\widetilde{\mathbf{M}}$, we obtain

$$\frac{\partial \boldsymbol{u}_i(\widetilde{\mathbf{M}})}{\partial \widetilde{\mathbf{M}}_{ij}} = \mathbf{0}; \quad \frac{\partial \tilde{\boldsymbol{u}}_i(\widetilde{\mathbf{M}})}{\partial \widetilde{\mathbf{M}}_{ij}} = \left(\lambda_U \mathbf{I}_k + \boldsymbol{\Sigma}_U^{(i)}\right)^{-1} \boldsymbol{v}_j;$$

$$\frac{\partial \boldsymbol{v}_j(\widetilde{\mathbf{M}})}{\partial \widetilde{\mathbf{M}}_{ij}} = \left(\lambda_V \mathbf{I}_k + \boldsymbol{\Sigma}_V^{(j)}\right)^{-1} \boldsymbol{u}_i.$$

Here $\boldsymbol{\Sigma}_U^{(i)}$ and $\boldsymbol{\Sigma}_V^{(j)}$ are defined as

$$\boldsymbol{\Sigma}_U^{(i)} = \sum_{j \in \Omega_i \cup \widetilde{\Omega}_i} \boldsymbol{v}_j \boldsymbol{v}_j^\top, \quad \boldsymbol{\Sigma}_V^{(j)} = \sum_{i \in \Omega'_j \cup \widetilde{\Omega}'_j} \boldsymbol{u}_i \boldsymbol{u}_i^\top. \tag{12}$$

A framework of the proposed optimization algorithm is described in Algorithm 1.

## 4.2 Attacking Nuclear Norm Minimization

We extend the projected gradient ascent algorithm in Sec. 4.1 to compute optimal attack strategies for the nuclear norm minimization formulation in Eq. (5). Since the objective in Eq. (5) is convex, the global optimal solution $\boldsymbol{\Theta} = (\mathbf{X}, \widetilde{\mathbf{X}})$ can be obtained by conventional convex optimization procedures such as proximal gradient descent (a.k.a. singular value thresholding [3] for nuclear norm minimization). In addition, the resulting estimation $(\mathbf{X}; \widetilde{\mathbf{X}})$ is low rank due to the nuclear norm penalty [2]. Suppose $(\mathbf{X}; \widetilde{\mathbf{X}})$ has rank $\rho \leq \min(m, n)$. We use $\boldsymbol{\Theta}' = (\mathbf{U}, \widetilde{\mathbf{U}}, \mathbf{V}, \boldsymbol{\Sigma})$ as an alternative characterization of the learnt model with a reduced number of parameters. Here $\mathbf{X} = \mathbf{U}\boldsymbol{\Sigma}\mathbf{V}^\top$ and $\widetilde{\mathbf{X}} = \widetilde{\mathbf{U}}\boldsymbol{\Sigma}\mathbf{V}^\top$ are singular value decompositions of $\mathbf{X}$ and $\widetilde{\mathbf{X}}$; that is, $\mathbf{U} \in \mathbb{R}^{m \times \rho}$, $\widetilde{\mathbf{U}} \in \mathbb{R}^{m' \times \rho}$, $\mathbf{V} \in \mathbb{R}^{n \times \rho}$ have orthornormal columns and $\boldsymbol{\Sigma} = \mathbf{diag}(\sigma_1, \cdots, \sigma_\rho)$ is a non-negative diagonal matrix.

To compute the gradient $\nabla_{\widetilde{\mathbf{M}}} R(\widehat{\mathbf{M}}, \mathbf{M})$, we again apply the chain rule to decompose the gradient into two parts:

$$\nabla_{\widetilde{\mathbf{M}}} R(\widehat{\mathbf{M}}, \mathbf{M}) = \nabla_{\widetilde{\mathbf{M}}} \boldsymbol{\Theta}'_\lambda(\widetilde{\mathbf{M}}; \mathbf{M}) \nabla_{\boldsymbol{\Theta}'} R(\widehat{\mathbf{M}}, \mathbf{M}). \tag{13}$$

Similar to Eq. (11), the second gradient term $\nabla_{\boldsymbol{\Theta}'} R(\widehat{\mathbf{M}}, \mathbf{M})$ is relatively easier to evaluate. Its derivation details are deferred to the Appendix. In the remainder of this section we shall focus on the computation of the first gradient term, which involves partial derivatives of $\boldsymbol{\Theta}' = (\mathbf{U}, \widetilde{\mathbf{U}}, \mathbf{V}, \boldsymbol{\Sigma})$ with respect to malicious users $\widetilde{\mathbf{M}}$.

We begin with the KKT condition at the optimal solution $\boldsymbol{\Theta}'$ of Eq. (5). Unlike the alternating minimization formulation, the nuclear norm function $\|\cdot\|_*$ is not everywhere differentiable. As a

---

**Algorithm 2** Optimizing $\widetilde{\mathbf{M}}$ via SGLD

---

1: **Input**: Original partially observed $m \times n$ data matrix $\mathbf{M}$, algorithm regularization parameter $\lambda$, attack budget parameters $\alpha$, $B$ and $\Lambda$, attacker's utility function $R$, step size $\{s_t\}_{t=1}^{\infty}$, tuning parameter $\beta$, number of SGLD iterations $T$.
2: **Prior setup**: compute $\xi_j = \frac{1}{m} \sum_{i=1}^{m} \mathbf{M}_{ij}$ and $\sigma_j^2 = \frac{1}{m} \sum_{i=1}^{m} (\mathbf{M}_{ij} - \xi_j)^2$ for every $j \in [n]$.
3: **Initialization**: sample $\widetilde{\mathbf{M}}_{ij}^{(0)} \sim \mathcal{N}(\xi_j, \sigma_j^2)$ for $i \in [m']$ and $j \in [n]$.
4: **for** $t = 0$ to $T$ **do**
5:     Compute the optimal solution $\boldsymbol{\Theta}_\lambda(\widetilde{\mathbf{M}}^{(t)}; \mathbf{M})$.
6:     Compute gradient $\nabla_{\widetilde{\mathbf{M}}} R(\widehat{\mathbf{M}}, \mathbf{M})$ using Eq. (10).
7:     Update $\widetilde{\mathbf{M}}^{(t+1)}$ according to Eq. (17).
8: **end for**
9: **Projection**: find $\widetilde{\mathbf{M}}^* \in \arg\min_{\widetilde{\mathbf{M}} \in \mathbb{M}} \|\widetilde{\mathbf{M}} - \widetilde{\mathbf{M}}^{(t)}\|_F^2$. Details in the main text.
10: **Output**: $m' \times n$ malicious matrix $\widetilde{\mathbf{M}}^*$.

---

result, the KKT condition relates the *subdifferential* of the nuclear norm function $\partial \| \cdot \|_*$ as

$$\mathcal{R}_{\Omega, \tilde{\Omega}}\left([\mathbf{M}; \widetilde{\mathbf{M}}] - [\mathbf{X}; \widetilde{\mathbf{X}}]\right) \in \lambda \partial \|[\mathbf{X}; \widetilde{\mathbf{X}}]\|_*. \tag{14}$$

Here $[\mathbf{X}; \widetilde{\mathbf{X}}]$ is the concatenated $(m + m') \times n$ matrix of $\mathbf{X}$ and $\widetilde{\mathbf{X}}$. The subdifferential of the nuclear norm function $\partial \| \cdot \|_*$ is also known [2]:

$$\partial \|\mathbf{X}\|_* = \left\{ \mathbf{U}\mathbf{V}^\top + \mathbf{W} : \mathbf{U}^\top \mathbf{W} = \mathbf{W}\mathbf{V} = \mathbf{0}, \|\mathbf{W}\|_2 \le 1 \right\},$$

where $\mathbf{X} = \mathbf{U}\boldsymbol{\Sigma}\mathbf{V}^\top$ is the singular value decomposition of $\mathbf{X}$. Suppose $\{\boldsymbol{u}_i\}$, $\{\tilde{\boldsymbol{u}}_i\}$ and $\{\boldsymbol{v}_j\}$ are rows of $\mathbf{U}, \widetilde{\mathbf{U}}, \mathbf{V}$ and $\mathbf{W} = \{w_{ij}\}$. We can then re-formulate the KKT condition Eq. (14) as follows:

$$\forall(i, j) \in \Omega, \quad \mathbf{M}_{ij} = \boldsymbol{u}_i^\top (\boldsymbol{\Sigma} + \lambda \mathbf{I}_\rho)\boldsymbol{v}_j + \lambda w_{ij};$$
$$\forall(i, j) \in \widetilde{\Omega}, \quad \widetilde{\mathbf{M}}_{ij} = \tilde{\boldsymbol{u}}_i^\top (\boldsymbol{\Sigma} + \lambda \mathbf{I}_\rho)\boldsymbol{v}_j + \lambda \tilde{w}_{ij}.$$

Now we derive $\nabla_{\widetilde{\mathbf{M}}} \boldsymbol{\Theta} = \nabla_{\widetilde{\mathbf{M}}}(\boldsymbol{u}, \tilde{\boldsymbol{u}}, \boldsymbol{v}, \sigma)$; the full derivation is deferred to the extended version [3].

## 4.3 Mimicking Normal User Behaviors

Normal users generally do not rate items uniformly at random. For example, some movies are significantly more popular than others. As a result, malicious users that pick rated movies uniformly at random can be easily identified by running a $t$-test against a known database consisting of only normal users, as shown in Sec. 5. To alleviate this issue, in this section we propose an alternative approach to compute data poisoning attacks such that the resulting malicious users $\widetilde{\mathbf{M}}$ mimics normal users $\mathbf{M}$ to avoid potential detection, while still achieving reasonably large utility $R(\widehat{\mathbf{M}}, \mathbf{M})$ for the attacker. We use a Bayesian formulation to take both data poisoning and detection avoidance objectives into consideration. The prior distribution $p_0(\widetilde{\mathbf{M}})$ captures normal user behaviors and is defined as a multivariate normal distribution

$$p_0(\widetilde{\mathbf{M}}) = \prod_{i=1}^{m'} \prod_{j=1}^{n} \mathcal{N}(\widetilde{\mathbf{M}}_{ij}; \xi_j, \sigma_j^2),$$

where $\xi_j$ and $\sigma_j^2$ are mean and variance parameters for the rating of the $j$th item provided by normal users. In practice both parameters can be estimated using normal user matrix $\mathbf{M}$ as $\xi_j = \frac{1}{m} \sum_{i=1}^{m} \mathbf{M}_{ij}$ and $\sigma^2 = \frac{1}{m} \sum_{i=1}^{m} (\mathbf{M}_{ij} - \xi_j)^2$. On the other hand, the likelihood $p(\mathbf{M}|\widetilde{\mathbf{M}})$ is defined as

$$p(\mathbf{M}|\widetilde{\mathbf{M}}) = \frac{1}{Z} \exp\left(\beta \cdot R(\widehat{\mathbf{M}}, \mathbf{M})\right), \tag{15}$$

where $R(\widehat{\mathbf{M}}, \mathbf{M}) = R(\widehat{\mathbf{M}}(\boldsymbol{\Theta}_\lambda(\widetilde{\mathbf{M}}; \mathbf{M})), \mathbf{M})$ is one of the attacker utility functions defined in Sec. 3, $Z$ is a normalization constant and $\beta > 0$ is a tuning parameter that trades off attack performance and

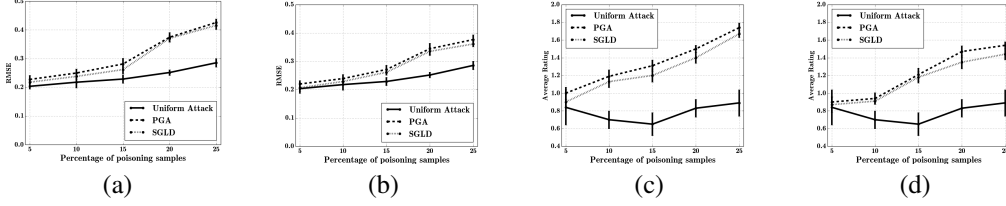

Figure 1: RMSE/Average ratings for alternating minimization with different percentage of malicious profiles; (a) $\mu_1 = 1, \mu_2 = 0$, (b) $\mu_1 = 1, \mu_2 = -1$, (c)$\mu_1 = 0, \mu_2 = 1$, (d)$\mu_1 = -1, \mu_2 = 1$.

detection avoidance. A small $\beta$ shifts the posterior of $\widetilde{\mathbf{M}}$ toward its prior, which makes the resulting attack strategy less effective but harder to detect, and vice versa.

Given both prior and likelihood functions, an effective detection-avoiding attack strategy $\widetilde{\mathbf{M}}$ can be obtained by sampling from its posterior distribution:

$$
p(\widetilde{\mathbf{M}}|\mathbf{M}) \quad = \quad p_0(\widetilde{\mathbf{M}})p(\mathbf{M}|\widetilde{\mathbf{M}})/p(\mathbf{M}) \quad \propto \quad \exp\left(-\sum_{i=1}^{m'}\sum_{j=1}^{n}\frac{(\widetilde{\mathbf{M}}_{ij} - \xi_j)^2}{2\sigma_j^2} + \beta R(\widehat{\mathbf{M}}, \mathbf{M})\right). \quad (16)
$$

Posterior sampling of Eq. (16) is clearly intractable due to the implicit and complicated dependency of the estimated matrix $\widehat{\mathbf{M}}$ on the malicious data $\widetilde{\mathbf{M}}$, that is, $\widehat{\mathbf{M}} = \widehat{\mathbf{M}}(\boldsymbol{\Theta}_\lambda(\widetilde{\mathbf{M}}; \mathbf{M})))$. To circumvent this problem, we apply *Stochastic Gradient Langevin Dynamics (SGLD, [10])* to approximately sample $\widetilde{\mathbf{M}}$ from its posterior distribution in Eq. (16). More specficly, the SGLD algorithm iteratively computes a sequence of posterior samples $\{\widetilde{\mathbf{M}}^{(t)}\}_{t \geq 0}$ and in iteration $t$ the new sample $\widetilde{\mathbf{M}}^{(t+1)}$ is computed as

$$
\widetilde{\mathbf{M}}^{(t+1)} = \widetilde{\mathbf{M}}^{(t)} + \frac{s_t}{2}\left(\nabla_{\widetilde{\mathbf{M}}}\log p(\widetilde{\mathbf{M}}|\mathbf{M})\right) + \varepsilon_t, \quad (17)
$$

where $\{s_t\}_{t \geq 0}$ are step sizes and $\varepsilon_t \sim \mathcal{N}(\mathbf{0}, s_t\mathbf{I})$ are independent Gaussian noises injected at each SGLD iteration. The gradient $\nabla_{\widetilde{\mathbf{M}}}\log p(\widetilde{\mathbf{M}}|\mathbf{M})$ can be computed as

$$
\nabla_{\widetilde{\mathbf{M}}}\log p(\widetilde{\mathbf{M}}|\mathbf{M}) = -(\widetilde{\mathbf{M}} - \boldsymbol{\Xi})\boldsymbol{\Sigma}^{-1} + \beta\nabla_{\widetilde{\mathbf{M}}}R(\widehat{\mathbf{M}}, \mathbf{M}),
$$

where $\boldsymbol{\Sigma} = \mathbf{diag}(\sigma_1^2, \cdots, \sigma_n^2)$ and $\boldsymbol{\Xi}$ is an $m' \times n$ matrix with $\boldsymbol{\Xi}_{ij} = \xi_j$ for $i \in [m']$ and $j \in [n]$. The other gradient $\nabla_{\widetilde{\mathbf{M}}}R(\widehat{\mathbf{M}}, \mathbf{M})$ can be computed using the procedure in Sections 4.1 and 4.2. Finally, the sampled malicious matrix $\widetilde{\mathbf{M}}^{(t)}$ is projected back onto the feasible set $\mathbb{M}$ by selecting $B$ items per user with the largest absolute rating and truncating ratings to the level of $\{\pm\Lambda\}$. A high-level description of the proposed method is given in Algorithm 2.

## 5 Experimental Results

To evaluate the effectiveness of our proposed poisoning attack strategy, we use the publicly available MovieLens dataset which contains 20 millions ratings and 465,000 tag applications applied to 27,000 movies by 138,000 users [23]. We shift the rating range to $[-2, 2]$ for computation convenience. To avoid the "cold-start" problem, we consider users who have rated at least 20 movies. Two metrics are employed to measure the relative performance of the systems before and after data poisoning attacks: root mean square error (RMSE) for the predicted unseen entries[4] and average rating for specific items. We then analyze the tradeoff between attack performance and detection avoidance, which is controled by the $\beta$ parameter in Eq. (15). This serves as a guide for how $\beta$ should be set in later experiments. We use a paired $t$-test to compare the distributions of rated items between normal and malicious users. We present the trend of p-value against different values of $\beta$ in the extended version of the paper. To strive for a good tradeoff, we set $\beta = 0.6$ at which the p-value stablizes around 0.7 and the poisoning attack performance is not significantly sacrificed.

We employ attack models specified in Eq. (9), where the utility parameters $\mu_1$ and $\mu_2$ balance two different malicious goals (availability and integrity) an attacker wishes to achieve. For the integrity

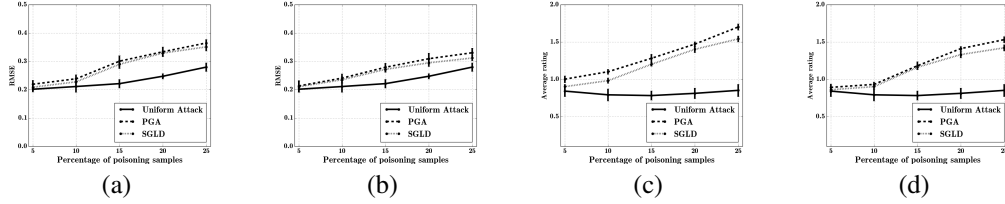

Figure 2: RMSE/Average ratings for nuclear norm minimization with different percentage of malicious profiles; (a) $\mu_1 = 1, \mu_2 = 0$, (b) $\mu_1 = 1, \mu_2 = -1$, (c)$\mu_1 = 0, \mu_2 = 1$, (d)$\mu_1 = -1, \mu_2 = 1$.

utility $R^{\text{in}}_{J_0,w}$, the $J_0$ set contains only one item $j_0$ selected randomly from all items whose average predicted ratings are around 0.8. The weight $w_{j_0}$ is set as $w_{j_0} = 2$. Figure 1 (a) (b) plots the RMSE after data poisoning attacks. When $\mu_1 = 1, \mu_2 = 0$, the attacker is interested in increasing the RMSE of the collaborative filtering system and hence reducing the system's availability. On the other hand, when $\mu_1 = 1, \mu_2 = -1$ the attacker wishes to increase RMSE while at the same time keeping the rating of specific items ($j_0$) as low as possible for certain malicious purposes. Figure 1 (b) shows that when the attackers consider to both objectives ($\mu_1 = 1, \mu_2 = -1$), the RMSE after poisoning is slightly lower than that if only availability is targeted ($\mu_1 = 1, \mu_2 = 0$). In addition, the *projected gradient ascent* (PGA) strategy generates the largest RMSE score compared with the other methods. However, PGA requires malicious users to rate each item uniformly at random, which might expose the malicious profiles to an informed defender. More specifically, the paired $t$-test on those malicious profiles produced by PGA rejects the null hypothesis that the items rated by the attacker strategies are the same as those obtained from normal users ($p < 0.05$). In contrast, the SGLD method leads to slightly worse attacker utility but generates malicious users that are hard to distinguish from the normal users (for example, the paired $t$-test leads to inconclusive p-values (larger than $0.7$) with $\beta = 0.6$. Finally, both PGA and SGLD result in higher attacker utility compared to *uniform attacks*, where both ratings and rated items are sampled uniformly at random for malicious profiles.

Apart from the RMSE scores, we also plot ratings of specific items against percentage of malicious profiles in Figure 1 (c) (d). We consider two additional attack utility settings: $\mu_1 = 0, \mu_2 = 1$, in which the attacker wishes to push the ratings of some particular items (specified in $w$ and $J_0$ of $R^{\text{in}}$) as high as possible; and $\mu_1 = -1, \mu_2 = 1$, where the attacker also wants to leave a "light trace" by reducing the impact on the entire system resulted from malicious activities. It is clear that targeted attackes (both PGA and SGLD) are indeed more effective at manipulating ratings of specific items for integrity attacks.

We also plot RMSE/Average ratings against malicious user percentage in Figure 2 for the nuclear norm minimization under similar settings based on a subset of 1000 users and 1700 movies (items), since it is more computationally expensive than alternating minimization. In general, we observe similar behavior of both RMSE/Average ratings under different attacking models $\mu_1, \mu_2$ with alternating minimization.

## 6   Discussion and Concluding Remarks

Our ultimate goal for the poisoning attack analysis is to develop possible defensive strategies based on the careful analysis of adversarial behaviors. Since the poisoning data is optimized based on the attacker's malicious objectives, the correlations among features within a feature vector may change to appear different from normal instances. Therefore, tracking and detecting deviations in the feature correlations and other accuracy metrics can be one potential defense. Additionally, defender can also apply the combinational models or sampling strategies, such as bagging, to reduce the influence of poisoning attacks.

## Acknowledgments

This research was partially supported by the NSF (CNS-1238959, IIS-1526860), ONR (N00014-15-1-2621), ARO (W911NF-16-1-0069), AFRL (FA8750-14-2-0180), Sandia National Laboratories, and Symantec Labs Graduate Research Fellowship.

## Footnotes

* Both authors contribute equally

[2]Note that when the collaborative filtering algorithm and its parameters are set, $\overline{\mathbf{M}}$ is a function of observed entries $\mathcal{R}_\Omega(\mathbf{M})$.

[3] http://arxiv.org/abs/1608.08182

[4]defined as RMSE $= \sqrt{\sum_{(i,j) \in \Omega^C}(\overline{\mathbf{M}}_{ij} - \widehat{\mathbf{M}}_{ij})^2/|\Omega^C|}$, where $\overline{\mathbf{M}}$ is the prediction of model trained on clean data $\mathcal{R}_\Omega(\mathbf{M})$ only (i.e., without data poisoning attacks).

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
