[Reviews · NeurIPS 2016]

Reviewer 1

Summary

The authors consider the problem of data poisoning attacks for matrix completing, which is a common problem in modern world. In this paper, an optimization algorithm based on PGA is proposed for attacking alternating minimization and nuclear norm minimization. In order to mimic normal user behaviors, another SGLD-based Bayesian algorithm is also proposed for the same problems. Experiments shows the effectiveness of proposed algorithms.

Qualitative Assessment

As a special area of machine learning, adversarial attacks for concrete learning algorithms draw lots of attentions these years. This paper give a detailed characterization for collaborative filtering, which I think is quite interesting and meaningful. Besides, the paper is well-written and easy to read. However, there are some concerns for the detailed content: 1. A basic assumption to implement attack algorithms is to know all users’ data, is this too strong in real world? In contrast, in most time, we only know a part of users’ information or some statistical information over all users, what conclusions can we obtain once in this situation? 2. In the process of computing optimal attack strategies, the algorithm has to random choose B items. I know this is an intuitive way to solve the non-convex domain constrain, but I suspect it may severely harm the effectiveness of its final solution. As I think adversaries may tend to choose some specific items they are interested in, random choosing items might lose many things. However from the experimental results’ view, it seems there is no such problem, so is there any explanation for this heuristics method? 3. For algorithm 1, I do not see any convergence analysis, so how could we ensure the algorithm will stop after some iterations and the utility of final solution? 4. Though the idea to mimic normal user behaviors is interesting, and seems novel in existing literature, overall, I feel that the main contribution of this paper is not very impressive. As utility functions for attackers are kind of straightforward in matrix completing problem, and the core idea to solve this type of objective function has also been used for several years.

Confidence in this Review

2-Confident (read it all; understood it all reasonably well)


Reviewer 2

Summary

The paper proposes a formal optimization framework for injecting fake ratings into collaborative filtering algorithms. The goal of the attacker is to 1) increase the error of the CF algorithm ("availability attack"), and 2) make the algorithm favor items that the attacker is interested in ("integrity attack"). Two common CF formulations are considered, alternating minimization and nuclear-norm minimization. The paper also discusses how to create realistically looking ratings that the system cannot distinguish easily from real ratings. The proposed method relies on stochastic gradient langevin dynamics to do approximate sampling of the posterior. The evaluation on the MovieLens dataset shows that the proposed method achieves a higher error than a uniform attack, while being slightly worse than a method that does not aim at creating realistically looking fake ratings.

Qualitative Assessment

The paper is very well-written and straightforward to follow. The attack formulation as an optimization problem is novel and interesting. A couple of suggestions for improvement, either in this paper or follow-up papers: 1) The assumption that the attacker knows the original data matrix M of the CF algorithm is very strong. It would be good to do some analysis on how the attacker can proceed under different possible views into M. 2) The evaluation is performed on a single dataset. Multiple datasets would substantiate the conclusions. The figure sizes can be improved - they are very small and hard to read. Some typos to correct: "mimicing," "20 millions ratings," "specfically."

Confidence in this Review

2-Confident (read it all; understood it all reasonably well)


Reviewer 3

Summary

This paper investigates how attackers can inject malicious data into the training dataset of a recommendation system, in order to either maximise the error of the system, or increase/decrease the popularity of a subset of certain items. Specifically, they design a projective gradient ascent method (PGA) that optimises the attack strategy. They also propose a technique based on stochastic gradient Langevin dynamics (SGLD) optimization that can generate malicious data mimicking normal users. Experimental results show that PGA and SGLD are more effective than strawman uniform attacks.

Qualitative Assessment

The paper explores an important topic – adversarial machine learning. While the paper contributes interesting results, it seems slightly lacking in novelty/depth. In general, the paper is well presented. Both the attack models and strategies are clearly derived and explained. It is indeed important to have this kind of analysis to fully understand the vulnerability of collaborative filtering schemes. New connections are made (to SGLD). A main concern revolves around novelty/impact. Researchers have achieved numerous attacks on learners now (with most key references cited appropriately in the paper), demonstrating that the vulnerability of learning to poisoning is fundamentally well understood. Attacks tend to be formulated in the same way, as an optimisation on the prediction score function (or learning map; or the composition of the two); combined with some form of gradient descent, leaving just the calculation of (sub)gradients. It is interesting to follow through with this (for a learner of choice) if the attack has some closed form solution that can be defended against, or in some way provides a surprising conclusion – not the case here. Formulating the different kinds of attacks on learners (availability, integrity, etc.) is not novel as a contribution. The paper should probably cite Barreno et al. (how secure is machine learning circa 2006 or so), or the papers that followed it up by the same authors. Attacking the iterative fitting process (not just a global solution) appears more novel. However the calculations for the attacks, while very involved and non-trivial are not novel in themselves. For example exploiting KKT conditions (line 165) has been done by e.g. Mei and Zhu (AISTATS 2015) in the context of attacking LDA. Mimicry attacks are well known e.g. Tan et al. (2002) were one of the earliest (applied to intrusion detection); Wagner & Soto (2002); Wang et al. (2006) the Anagram system. Formulating them as an optimisation is neat. The hybrid attack doesn't seem as well motivated as the pure strategies. A point missing is a discussion on the feasibility of the attacks. Even if the attacker has complete knowledge of the target system, the overhead of computing the gradient does not seem to be trivial. In addition, the experimental results suggest that when the purpose is to maximise the system error, there is not much difference between PGA/SGLD and uniform attacks, until the percentage of poisoning samples reaches 10% or even 15% (Fig. 1 (a) (b), Fig. 2 (a) (b)). It isn't clear that the overhead (in computation, but more importantly sophistication) outweighs the gain. In practice, is it practical for attackers to insert that much malicious data? This is important, in the context of the next comment: that the uniform attack is a strawman. Another problem with the experiment is that considering uniform attacks, both ratings and items are randomly chosen. This doesn’t make sense when the purpose is to boost/reduce the popularity of certain specific items – would a realistic baseline not bias sampling towards such items (or perform something like a dictionary attack). Updating from author feedback: Thankyou for the response. While my position on novelty remains substantively the same, I would agree with the authors that the typical stance in cybersecurity is that worst-case attacks are indeed of interest. For one, "no security through obscurity" (we cannot guarantee robustness by assuming the attacker is unaware of training data) but also, the worst-case bounds the average case. That said, the most powerful attacks are those that demonstrate influence over a target system while leveraging minimal capabilities.

Confidence in this Review

3-Expert (read the paper in detail, know the area, quite certain of my opinion)


Reviewer 4

Summary

The paper designs methods to attack matrix factorization based recommender systems given that the adversary has the ability to poison training data. Three different (actually two and a combination of both) adversarial objectives are defined. The paper then goes further to discuss hiding adversarial's identity (mimicking normal user) by adjusting how much the adversary pushes to realize its objective. Experiments performed to show the effectiveness of the approaches.

Qualitative Assessment

This paper is very well written and organized. A joy to read. Technical quality: Ideas are clearly presented; assumptions explicitly stated; KKT condition explained and examined. Novelty/Originality: There has been paper examining various other models' robustness to data poisoning, such as SVMs, LRs and neural nets. This paper is definitely a good addition to the discussion and adds to the variety. What I find novel is the attacker's ability to mimick normal user. The paper has discussed the trade-off between hacking the system v.s. looking normal (assuming the user ratings following a normal distribution.) Very good practical concern. Potential Impact: The security setting in the paper is practical and important. Moreoever, this paper makes a step towards security game between learner and attacker. That being said, there's still a large gap to fill between the current model and an optimized learner-attack game, which is likely to be very hard. The paper can be enhanced if it discusses more defensive models such as detecting correlation between users, or consider an online setting in which the attacker tries to fool an established yet updating system. I'm looking forward to a next paper addressing more challenging settings! Some minor technical doubts on the paper: 1) Both Algorithm 1 and 2 refer to compute gradient using Eq (10). However Eq (10) just uses the gradient to update M_{t+1} without mention how the gradient is calculated. Is that a typo/wrong reference? 2)I'm curious about how many repeated experiments are performed for the result in Figure 1, as SGLD is a heuristic.

Confidence in this Review

2-Confident (read it all; understood it all reasonably well)


Reviewer 5

Summary

The paper studies the effect of a data poisoning attack(specifically in the case of dummy users submitting malicious ratings) in the context of a collaborative recommendation system. It suggests defense for two of the collaborative filtering algorithms - alternative minimization and nuclear norm minimization. The problem is framed as a matrix optimization problem ( different kinds of attacks lead to different optimization functions) and projective gradient ascent method is used to solve the optimization problem for alternative minimization approach. A Bayesian formulation is suggested for the problem on mimicking normal users. Experiments are performed on the movie lens dataset with RMSE metric performance compared with and without poisoning attack.

Qualitative Assessment

The paper studies an important problem specifically regarding malicious users in a recommendation systems and suggests interesting techniques to combat such attacks. The overall problem seems to be well formulated and the model seems to capture the nature of malicious attackers quite nicely. The results in Section 5 on experiments does not seem to be conveyed well.

Confidence in this Review

1-Less confident (might not have understood significant parts)


Reviewer 6

Summary

The paper examines ways to introduce data poisoning attacks in order to mess up the predictions of collaborative filtering systems. The approach assumes that the attacker knows completely the algorithm used by the system and specifically examines two popular such algorithms, alternative minimization and nuclear norm minimization. From the perspective of the attacker, the attacker may have different objectives such as maximizing the total prediction error made by the algorithm ("availability attack") as well as item specific errors ("integrity attack") and hybrids of these two. The authors use projective gradient descent methods to solve the respective optimization problems from the perspective of the attacker where the key technical problem is to approximately compute gradients exploiting KKT conditions. The paper also examines a different framework where the attacker tries to mask his entries so that they look similar to "real" data and provide a stochastic gradient Langevin dynamics approach to solve that problem. The paper concludes by running experiments on the MovieLens database.

Qualitative Assessment

The issues related to adversarial attacks to collaborative filtering systems are interesting, however, I think the particular approach has several drawbacks. It assumes perfect knowledge of the filtering algorithm to the attackers, which is not a realistic and critically it stops short from giving any defensive tactics on the side of the system designer. In fact, a key attribute of this ecosystem is that it is in a sense a zero-sum game where both parties try to outsmart each other. This critical component is not captured in the current model. I believe that the paper here makes a reasonable start in analyzing this setting, however, I believe that it does not provide compelling enough insights and hence should probably not be accepted to NIPS. Another way to make the model more practical would be to reduce the knowledge assumed by the attacker. How does the attacker deal with this model uncertainty?

Confidence in this Review

2-Confident (read it all; understood it all reasonably well)